Comparison of machine learning techniques to handle imbalanced COVID-19 CBC datasets

http://orcid.org/0000-0001-8534-3480 Dorn Marcio 1 2 3 mdorn@inf.ufrgs.br
http://orcid.org/0000-0003-4083-5881 Grisci Bruno Iochins 1
Narloch Pedro Henrique 1
http://orcid.org/0000-0002-2825-8295 Feltes Bruno César 1 4
Avila Eduardo 3 5
http://orcid.org/0000-0002-7556-7904 Kahmann Alessandro 6
Alho Clarice Sampaio 3 5
1 Institute of Informatics, Federal University of Rio Grande do Sul , Porto Alegre, RS , Brazil
2 Center of Biotechnology, Federal University of Rio Grande do Sul , Porto Alegre, RS , Brazil
3 Forensic Science, National Institute of Science and Technology , Porto Alegre, RS , Brazil
4 Department of Genetics, Federal University of Rio Grande do Sul , Porto Alegre, RS , Brazil
5 School of Health and Life Sciences, Pontifical Catholic University of Rio Grande do Sul , Porto Alegre, RS , Brazil
6 Institute of Mathematics, Statistics and Physics, Federal University of Rio Grande , Rio Grande, RS , Brazil
Nakaya Helder
Electronic publication date: 2021 Aug 12
Publication date: 2021
Volume: 7
Electronic Location ID: e670
Received 2021 Apr 22; Accepted 2021 Jul 20
Copyright: © 2021 Dorn et al.
Copyright year: 2021
Copyright holder: Dorn et al.
License: This is an open access article distributed under the terms of the Creative Commons Attribution License, which permits unrestricted use, distribution, reproduction and adaptation in any medium and for any purpose provided that it is properly attributed. For attribution, the original author(s), title, publication source (PeerJ Computer Science) and either DOI or URL of the article must be cited.
License URL: https://creativecommons.org/licenses/by/4.0/

Keywords: Machine learning, Data mining, Imbalanced datasets, Covid, Hemogram

Funding: Fundacao de Amparo a Pesquisa do Estado do Rio Grande do Sul - FAPERGS 19/2551-0001906-8 Conselho Nacional de Desenvolvimento Cientifico e Tecnologico - CNPq 311611/2018-4 Coordenacao de Aperfeicoamento de Pessoal de Nivel Superior - STICAMSUD 88881.522073/2020-01 DAAD/CAPES PROBRAL 88881.198766/2018-01 Coordenação de Aperfeiçoamento de Pessoal de Nivel Superior - Brasil (CAPES) 001 This work was supported by grants from the Fundacao de Amparo a Pesquisa do Estado do Rio Grande do Sul - FAPERGS [19/2551-0001906-8], Conselho Nacional de Desenvolvimento Cientifico e Tecnologico - CNPq [311611/2018-4], and the Coordenacao de Aperfeicoamento de Pessoal de Nivel Superior - STICAMSUD [88881.522073/2020-01] and DAAD/CAPES PROBRAL [88881.198766/2018-01]. This study was financed by the Coordenação de Aperfeiçoamento de Pessoal de Nivel Superior - Brasil (CAPES) - Finance Code 001. There was no additional external funding received for this study. The funders had no role in study design, data collection and analysis, decision to publish, or preparation of the manuscript.

==============================
The Coronavirus pandemic caused by the novel SARS-CoV-2 has significantly impacted human health and the economy, especially in countries struggling with financial resources for medical testing and treatment, such as Brazil’s case, the third most affected country by the pandemic. In this scenario, machine learning techniques have been heavily employed to analyze different types of medical data, and aid decision making, offering a low-cost alternative. Due to the urgency to fight the pandemic, a massive amount of works are applying machine learning approaches to clinical data, including complete blood count (CBC) tests, which are among the most widely available medical tests. In this work, we review the most employed machine learning classifiers for CBC data, together with popular sampling methods to deal with the class imbalance. Additionally, we describe and critically analyze three publicly available Brazilian COVID-19 CBC datasets and evaluate the performance of eight classifiers and five sampling techniques on the selected datasets. Our work provides a panorama of which classifier and sampling methods provide the best results for different relevant metrics and discuss their impact on future analyses. The metrics and algorithms are introduced in a way to aid newcomers to the field. Finally, the panorama discussed here can significantly benefit the comparison of the results of new ML algorithms.

Introduction

The Coronavirus disease (COVID-19) caused by the novel SARS-CoV-2 has spread from China and quickly transmitted to other countries. Since the beginning of 2020, the COVID-19 pandemic has significantly impacted human health and severely affected the global economy and financial markets (Nicola et al., 2020; Pak et al., 2020), especially in countries that cannot test their population and develop strategies to manage the crisis. In a scenario of large numbers of asymptomatic patients and shortages of tests, targeted testing is essential within the population (Peeling et al., 2020). The objective is to identify people whose immunity can be demonstrated and allow their safe return to their routine.

The diagnosis of COVID-19 is based on the clinical and epidemiological history of the patient (Ge et al., 2020) and the findings of complementary tests, such as chest tomography (CT-scan) (Bernheim et al., 2020; Ding et al., 2020) or nucleic acid testing (Kumar et al., 2020; Caruana et al., 2020). Nevertheless, the symptoms expressed by COVID-19 patients are nonspecific and cannot be used for an accurate diagnosis. CT-scan findings are seen with numerous pathogens and do not necessarily add diagnostic value (Gietema et al., 2020; Hope et al., 2020). Currently, Real-Time Polymerase Chain Reaction (RT-PCR) tests of viral RNA in fluid, typically obtained from the nasopharynx or oropharynx, are the gold-standard test for COVID-19 detection (Hadaya, Schumm & Livingston, 2020; Carter et al., 2020), together with proper clinical observations. The World Health Organization released several RT-PCR protocols to provide a proper diagnosis, help testing populations, and monitor the disease spread. However, using RT-PCR to diagnose COVID-19 has some limitations: reported sensitivities vary (Xu et al., 2020; Vogels et al., 2020); long turn-around times; and tests are not universally available (shortage of PCR primers, reagents or equipment) (Giri & Rana, 2020; Dhabaan, Al-Soneidar & Al-Hebshi, 2020).

The high demand for RT-PCR tests is highlighting the limitations of this type of diagnosis. Testing the entire population for COVID-19 is not feasible due to the cost, unavailability of PCR primers, lack of human and material resources, or even the delay from sample collection to test results. Instead, we need more targeted testing to manage the pandemic (Pulia et al., 2020; Eberhardt, Breuckmann & Eberhardt, 2020), and various efforts are being made worldwide to build strategies for such approach (Fang, 2020; Sheridan, 2020; Treibel et al., 2020; Zame et al., 2020). The optimal approach would be to collect and combine different data sources and use them to identify and prioritize the patients to be tested by RT-PCR. In this sense, complete blood count (CBC) is the world’s most widely available hematological laboratory test, where and (Ferrari et al., 2020) suggest routine blood tests as a potential diagnostic tool for COVID-19.

Moreover, hematological changes in patients affected by COVID-19 were reported in many works (Terpos et al., 2020; Lippi & Plebani, 2020; Han et al., 2020; Henry et al., 2020). Laboratory findings include leukopenia (Fan et al., 2020; Guan et al., 2020), lymphopenia (Guan et al., 2020; Bhatraju et al., 2020; Huang, Kovalic & Graber, 2020), and thrombocytopenia (Chen et al., 2020; Lippi, Plebani & Henry, 2020). Some authors have also suggested changes in the neutrophil/lymphocyte ratio in the severe disease progression of COVID-19 patients (Qu et al., 2020). However, defining the specific hematological alteration profile of COVID-19 differentiating it from other inflammatory or infectious processes is not simple.

Recently, artificial intelligence techniques, especially Machine Learning (ML), have been employed to analyze CBC data and assist in screening of patients with suspected COVID-19 infection (Yan et al., 2020; Yao et al., 2020; Gong et al., 2020; Alimadadi et al., 2020; Avila et al., 2020; Brinati et al., 2020a; Imran et al., 2020; Wu et al., 2020). ML is a huge field of study in Computer Science and Statistics that executes computational tasks through algorithms that rely on learning patterns from data samples to automate inferences. Class imbalance is common in many real-world applications and affects the quality and reliability of ML approaches (Leevy et al., 2018; Johnson & Khoshgoftaar, 2019; López et al., 2013). Most importantly, class imbalance is the reality of almost all biological datasets, as we demonstrated in previous works after the manual curation of more than 30.000 cancer datasets (Feltes et al., 2019; Feltes, Poloni & Dorn, 2021; Feltes et al., 2020). Imbalanced data refers to classification problems where we have an unequal number of instances for different classes. A well-known class imbalance scenario is the medical diagnosis task of detecting disease, where the majority of the patients are healthy, and the prediction of rare conditions is crucial (Katsanis et al., 2018). Additionally, it is common for biological datasets to be imbalanced since there are numerous limitations in generating, managing, and acquiring new samples, especially clinical data that heavily depends on patients willing to release their data or participating in clinical trials. Learning from these imbalanced data sets can be difficult, and non-standard ML methods are often required to achieve desirable results, especially in situations of low-prevalence diseases or clinical conditions.

In ML, a major issue is the release of multiple approaches, all valid in their way, but that needs to be discussed to provide a proper panorama of their applications on different types of data. Additionally, due to its low-cost nature, applying ML approaches to aid medical decision making is invaluable for countries struggling with financial resources to make strategic medical decisions.

This paper aims to: (i) review predictive ML techniques to predict the positivity or negativity for COVID-19 from CBC data; (ii) evaluate the impact of eight different classifiers and five distinct sampling methods already used for CBC data on three Brazilian CBC datasets; and (iii) evaluate which is the best overall classifier, as well as for each particular case.

In this sense, the eight classifiers were Support Vector Machines (SVM), Decision Trees (DT), K-Nearest Neighbors (KNN), Random Forest (RF), Multi-Layer Perceptron (MLP), Logistic Regression (LR), Naïve Bayes (NB), and eXtreme Gradient Boosting (XGBoost). Moreover, the five tested sampling methods for the imbalanced class problem were Random Under Sampling (RUS), Random Over Sampling (ROS), Adaptive Synthetic Sampling (ADASYN), Synthetic Minority Over Sampling TEchnique (SMOTE), and Synthetic Minority Over Sampling TEchnique Tomek links (SMOTETomek). Considering the importance of the application, the number of different algorithms available, and the rapid increase in publications reporting different ML approaches to handle COVID-19 CBC data, a survey summarising the main advantages, drawbacks, and challenges of the field can significantly aid future works. A workflow summarizing the steps taken in this work can be found in Fig. 1.

Figure 1 Methodological steps used in this work.

The survey will first explain the employed methodology, the tested datasets’ characteristics, and the chosen evaluation metrics. Afterward, a brief review of the major ML predictors used on CBC COVID-19 datasets is conducted, followed by a review of techniques to handle imbalanced data. This exposition is succeeded by describing the main findings, listing the lessons learned from the survey, and conclusions.

Preliminaries

Datasets

At the time of this work, Brazil was the third country most affected by the COVID-19 pandemic, reaching more than 18 million confirmed cases. Thus, discussing data gathered from Brazil can become invaluable to understand SARS-CoV-2 data. Complete datasets used in the present study were obtained from an open repository of COVID-19-related cases in Brazil. The database is part of the COVID-19 Data Sharing/BR initiative (Mello et al., 2020), and it is comprised of information about approximately 177,000 clinical cases. Patient data were collected from three distinct private health services providers in the São Paulo State, namely the Fleury Group (https://www.fleury.com.br), the Albert Einstein Hospital (https://www.einstein.br) and the Sírio-Libanês Hospital (https://www.hospitalsiriolibanes.org.br), and a database for patients from each institution was built. The data from COVID-19 patients was collected from February 26th, 2020 to June 30th, 2020, and the control data (individuals without COVID-19) was collected from November 1st, 2019 to June 30th, 2020.

Patient data is provided in an anonymized form. Three distinct types of patients information are provided in this repository: (i) patients demographic data (including sex, year of birth, and residence zip code); (ii) clinical and/or laboratory exams results (including different combinations of the following data: hemogram and blood cell count results, blood tests for a biochemical profile, pulmonary function tests, and blood gas analysis, diverse urinalysis parameters, detection of a panel of different infectious diseases, pulmonary imaging results (X-ray or CT scans), among others. COVID-19 detection by RT-PCR tests is described for all patients, and serology diagnosis (in the form of specific IgG and IgM antibody detection) is provided for some samples; and (iii) when available, information on each patient clinical progression and transfers, hospitalization history, as well as the disease outcome (primary endpoints, as death or recuperation). Available information is not complete for all patients, with a distinct combination of results provided individually.

Overall baseline characteristics can be found on the complete database, available at the FAPESP COVID-19 Data Sharing/BR (https://repositoriodatasharingfapesp.uspdigital.usp.br/). The most common clinical test results available for all patients is the hemogram data. As such, it was selected for the testing of the current sample set. Twenty distinct hemogram test parameters were obtained from the database, including hematocrit (%), hemoglobin (g/dl), platelets (×103 μl), mean platelet volume (fl), red blood cells (×106 μl), lymphocytes (×103 μl), leukocytes (×103 μl), basophils (×103 μl), eosinophils (×103 μl), monocytes (×103 μl), neutrophils (×103 μl), mean corpuscular volume (MCV) (fl), mean corpuscular hemoglobin (MCH) (pg), mean corpuscular hemoglobin concentration (MCHC) (g/dl), red blood cell distribution width (RDW) (%), % Basophils, % Eosinophils, % Lymphocytes % Monocytes, and % Neutrophils (Figs. 1 and 2).

Figure 2 Distributions of white blood cells related variables for positive (purple) and negative (green) classes of the three datasets: Albert Einstein Hospital (HAE), Fleury Group (FLE), and Sírio-Libanês Hospital (HSL). The central white dot is the median.

Patients with incomplete (missing data) or no data available for the above parameters were not included in the present analysis. For patients with more than a single test result available, a unique hemogram test was used, with the selection based on the blood test date. In this sense, same-day results to the PCR-test collection date was adopted as a reference, or the day closest to the test.

More information regarding the three distinct datasets’ distributions can be found in Figs. 2 and 3. The most relevant information assessed in the present study is database size, the number of available clinical test results, gender distribution (male or female), and COVID-19 RT-PCR test result (classified as positive or negative) ratio. The parameters for each data subset are described for the original dataset and for the subset of selected samples used in this study (after removal of patients containing missing values), as seen in Table 1.

Figure 3 Distributions of red blood cells related variables for positive (purple) and negative (green) classes of the three datasets: Albert Einstein Hospital (HAE), Fleury Group (FLE), and Sírio-Libanês Hospital (HSL). The central white dot is the median.

Table 1 Data Summary of the initial full dataset and selected subsets of samples.

Albert Einstein Hospital (HAE); Fleury Group (FLE) and Sírio-Libanês Hospital (HSL). Class ratio is represented as the ratio of the total of selected positive/negative samples.

Dataset	Samples	PCR Positive	PCR Negative	Class ratio	
	Original	Selected	Male	Female	Total	Male	Female	Total		
HAE	44,879	4,567	758	642	1,400	1,461	1,706	3,167	0.442	
FLE	129,597	803	111	145	256	225	322	547	0.468	
HSL	2,732	515	301	202	503	9	3	12	41.916	
Total samples	177,208	5,885	1,170	989	2,159	1,695	2,031	3,726	0.579	

The column “class ratio” in Table 1 shows the level of class imbalance for each dataset. It was computed by dividing the number of positive samples by the number of negative samples. The number of negative samples from the Albert Einstein Hospital and the Fleury Group exceeds the positive samples. This is expected from disease data since the number of infections will be small compared to the entire population. However, in the Sírio-Libanês Hospital data, there is over forty times the amount of positive samples compared to negative samples. This represents another source of bias in the data acquisition: the dataset consists of patients tested because they had already shown COVID-19-like symptoms, skewing the data to positive samples. This is crucial because the decision to test a patient for COVID-19 in institutions that struggle with funds is a common judgment call. Datasets with an apparent biased disease prevalence, as is the case with the Sírio-Libanês Hospital data (in reality the positive class for COVID-19 is not expected to be 40 times more prevalent), should be discarded from biological analysis. These datasets are being critically evaluated and used only as examples for this research. The last columns of Table 1 also show that, in general, the variables do not belong to the same distribution for the three centers, regardless of the classes.

Data characterization

For data characterization, we use two metrics: the Bhattacharyya Distance (BD); and the Kolmogorov–Smirnov statistics (KS). We will now present both metrics, followed by a discussion of its results in the studied datasets. The goal is to determine the separability between the negative and positive classes among the three datasets. BD calculates the separability between two Gaussian distributions (Anzanello et al., 2015). However, it depends on the covariance inverse matrix for multivariate cases, which can be nonviable for datasets with high dimensionality, such as the ones employed in this paper. Therefore, we will use its univariate form as in Eq. (1) (Coleman & Andrews, 1979).

(1) Bj(b,s)=14+ln(14(σbj2σsj2+σsj2σbj2+2))+14((υbj−υsj)2σbj2+σsj2)

where σ2 and υ are the variance and mean of the statistical distributions of the j − th variable for groups b and s, respectively. The first part of Eq. (1) distinguishes classes by the differences between variances, while the second part distinguishes classes by the differences between its weighted means. For classification purposes, we would expect low variance within classes and a high difference between means. Therefore, we will complement the BD value by analyzing the probability density functions to verify which part of Eq. (1) influences the highest BD values.

The other employed characterization metric is the D statistic from the two samples Kolmogorov–Smirnov test (KS test). The KS test is a non-parametric approach that quantifies the maximum difference between samples’ univariate empirical cumulative distribution values (i.e., the maximum separability between two distributions) (Kahmann et al., 2018) (Eq. 2).

(2) Dw=maxx⁡(|F1(x)−F2(x)|)

where D is the D statistic, such that w denotes which hemogram result is being analyzed, F1 and F2 are the cumulative empirical distributions of classes 1 and 2, and x are the obtained hemogram result. Dw values belongs to the [0,1] interval, where values closer to one suggest higher separability between classes (Xiao, 2017). Table 2 shows the D statistics and BD for all variables for the three datasets. Firstly, we will discuss the BD and D statistic results for each dataset, followed by comparing such results among all datasets.

Table 2 Separability between the negative and positive classes among the three datasets: Albert Einstein Hospital (HAE), Fleury Group (FLE), and Sírio-Libanês Hospital (HSL).

The measurements use the D statistic from the two samples Kolmogorov–Smirnov test and the Bhattacharyya Distance (BD). Results discussed in the main text are in bold. The last two columns show the Kruskal–Wallis H test (KW) together with its p-value, to compare the variables distributions for the three centers, regardless of the outcome. In this case, results rejecting the Null Hypothesis that data belongs to the same distribution are in bold.

Dataset	HAE	Fleury	HSL		
Metric	D	BD	D	BD	D	BD	KW	p-value	
Basophils	0.443474	0.1093475	0.398323	0.1239094	0.55301	0.1287230	117.02	0.0	
Basophils#	0.261855	0.0444302	0.266960	0.0726225	0.51341	0.0608966	110.27	0.0	
Eosinophils	0.364556	0.1221014	0.375792	0.0251153	0.40357	0.0643121	81.37	0.0	
Eosinophils#	0.277567	0.0796526	0.291354	0.0214845	0.25447	0.0534425	64.08	0.0	
Hematocrit	0.046150	0.0013166	0.066184	0.0009195	0.21868	0.1026450	69.75	0.0	
Hemoglobin	0.044774	0.0013781	0.046446	0.0008058	0.22465	0.0750943	29.04	0.0	
Leukocytes	0.333118	0.0479674	0.264275	0.0523979	0.38386	0.0390140	197.68	0.0	
Lymphocytes	0.369636	0.0627853	0.255698	0.0615829	0.37673	0.4035528	141.34	0.0	
Lymphocytes#	0.120357	0.0053703	0.079417	0.0048503	0.21189	0.0285911	187.50	0.0	
MCH	0.040136	0.0014844	0.129213	0.0033433	0.13320	0.0467845	7.46	0.024	
MCHC	0.061690	0.0023295	0.087915	0.0007333	0.24138	0.0265250	511.26	0.0	
MCV	0.025520	0.0011544	0.112824	0.0033957	0.12624	0.0055598	118.32	0.0	
MPV	0.085494	0.0044252	0.103647	0.0019528	0.39413	0.0611329	77.42	0.0	
Monocytes	0.137388	0.0097614	0.084245	0.0007295	0.50712	0.2983040	68.42	0.0	
Monocytes#	0.212097	0.0395842	0.253870	0.0659478	0.28661	0.0475931	26.77	0.0	
Neutrophils	0.208235	0.0170758	0.207523	0.0287769	0.21471	0.0171623	208.71	0.0	
Neutrophils#	0.102965	0.0041523	0.115902	0.0060480	0.24121	0.0357412	175.81	0.0	
Platelets	0.198826	0.0170292	0.254356	0.0257660	0.13800	0.0069387	39.01	0.0	
RDW	0.054032	0.0009597	0.050259	0.0017099	0.23707	0.0539648	4.28	0.1171	
RedbloodCells	0.037971	0.0010371	0.075089	0.0030830	0.16186	0.0401366	30.13	0.0	

Regarding the dataset separability in the HSL dataset, the D statistic yields Basophils, Basophils#, Monocytes, and Eosinophils as the variables with higher distance between the Cumulative Probability Function from positive and negative diagnosed patients. Complementing this analysis by the BD and the Probability Density Function (PDF) represented in Fig. 2, the distribution of Basophils, Basophils#, and Eosinophils from the negative patients has a higher mean. Besides the higher D statistics, the BD is lower than other variables, indicating that the distributions are similar; however, one group (in this case, the negative group) has systematically higher values. On the other hand, the other variable with high D statistic (Monocytes) has a flattened distribution for the negative patients, increasing its variance and consequentially its BD once the positive cases variance is small. The small sample size may jeopardize such distribution for negative patients. Complementarily, it is notable that this variable does not have a linear separation between classes.

As for the HAE dataset, the variables Basophils, Lymphocytes, Eosinophils and Leukocytes yields the higher D statistic. All of them have the same characteristic: similar distribution but with negative distribution with higher values. It is noteworthy that the higher BD (Basophils and Eosinophils) can be attributed to outliers. From Figs. 2 and 3 it can be noticed that such distributions present different means (as corroborated by the D statistic) combined with spurious values with high distance from the modal distribution point, resulting in a larger variance.

The variables yielding the higher D statistic on the FLEURY dataset are Basophils and Eosinophils. Regarding the Basophils distributions, the curve from negative cases is flatterer than the positive case curve. Even so, it is notable that the negative distribution has higher values, and both variances are small, resulting in a high BD. For the comparison of the Eosinophils in positive and negative cases, the existence of spurious values increases the variance for both distributions. However, the D statistic indicates that this variable provides good separability between classes.

Moreover, besides having variables with more potential separability (D statistics), the imbalance between classes is much more significant on the HSL dataset, which may bias such analysis. Both HAE and Fleury datasets have similar characteristics regarding classes’ sample size proportions. However, the HAE has more variables with high D statistics, and its values are higher as well.

On a final note, CBC data is highly prone to fluctuations. Some variables, such as age and sex, are among the most discussed sources of immunological difference, but others are sometimes unaccounted. For example, a systematic review in 2015 by Paynter et al. (2015) demonstrated that the immune system is significantly modulated by distinct seasonal changes in different countries, which, by its turn, impact respiratory and infectious diseases. Similarly, circadian rhythm can also impact the circulation levels of different leukocytes (Pritchett & Reddy, 2015). Distinct countries have specific seasonal fluctuations and, sometimes, extreme circadian regulations - thus, immune responses’ inherent sensibility should always be considered a potential bias. This also impacts the comparison between different computational approaches that use datasets from other researchers for testing or training. While this work focuses on the application of ML and sampling algorithms to this data, a more in-depth biological analysis regarding the interaction between sex, age, and systemic inflammation from these Brazilian datasets can be found in the work of Ten-Caten et al. (2021).

Evaluation metrics

The metrics to evaluate how well a classifier performs in discriminating between the target condition (positive for COVID-19) and health can be derived from a “confusion matrix” (Table 3) that contrasts the “true” labels obtained from the “gold standard” to the predicted labels. From it, we have four possible outcomes: either the classifier correctly assigns a sample as positive (with the target condition) or as negative (without the target condition), and in this case, we have true positives, and true negatives or the prediction is wrong, leading to false positives or false negatives.

Table 3 Confusion matrix of binary classification.

		“Gold standard”	
		Subjects with the disease	Subjects without the disease	
Classifier	Predicted as positive	TP	FP (Type I Error)	
	Predicted as negative	FN (Type II Error)	TN	
Note:

TP, True positives; TN, True negatives; FP, False positives; FN, False negatives.

Some metrics can assess the discriminative property of the test, while others can determine its predictive ability (Šimundić, 2009), and not all are well suited for diagnostic tasks because of imbalanced data (Tharwat, 2020). For instance, accuracy, sometimes also referred to as diagnostic effectiveness, is one of the most used classification performance (Tharwat, 2020). Still, it is greatly affected by the disease prevalence, and increases as the disease prevalence decrease (Šimundić, 2009). Overall, prediction metrics alone won’t reflect the biological meaning of the results. Consequentially, especially in diagnostic tasks, ML approaches should always be accompanied by expert decisions on the final results.

This review focuses on seven distinct metrics commonly used in classification and diagnostic tasks that are well suited for imbalanced data (Šimundić, 2009; Tharwat, 2020). This also allows for a more straightforward comparison of results in the literature. Each of these metrics evaluates a different aspect of the predictions and is listed in Table 4 together with a formula on how they can be computed from the results of the confusion matrix.

Table 4 Metrics used to compare the algorithms.

Metric	Formula	Range	Target value	
Sensitivity	TP/(TP + FN)	[0, 1]	∼1	
Specificity	TN/(FP + TN)	[0, 1]	∼1	
LR+	Sensitivity/(1-specificity)	[0, +∞)	>10	
LR−	(1-sensitivity)/specificity	[0, +∞)	<0.1	
DOR	(TP/FN)/(FP/TN)	[0, +∞)	>1	
F1-score	TP/(TP + 1/2 (FP + FN))	[0, 1]	∼1	
AUROC	Area under the ROC curve	[0, 1]	∼1	
Note:

TP, True positives; TN, True negatives; FP, False positives; FN, False negatives.

Sensitivity (also known as “recall”) is the proportion of correctly positive classified samples among all positive samples. It can be understood as the probability of getting a positive prediction in subjects with the disease or a model’s ability to recognize samples from patients (or subjects) with the disease. Analogously, specificity is the proportion of correctly classified negative samples among all negative samples, describing how well the model identifies subjects without the disease. Sensitivity and specificity are not dependants on the disease prevalence in examined groups (Šimundić, 2009).

The likelihood ratio (LR) is a combination of sensitivity and specificity used in diagnostic tests. The ratio of the expected test results in samples from patients (or subjects) with the disease to the samples without the disease. LR+ measures how much more likely it is to get a positive test result in samples with the disease than samples without the disease, and thus, it is a good indicator for ruling-in diagnosis. Good diagnostic tests usually have an LR+ larger than 10 (Šimundić, 2009). Similarly, LR- measures how much less likely it is to get a negative test result in samples with the disease when compared to samples without the disease, being used as an indicator for ruling-out the diagnosis. A good diagnostic test should have an LR- smaller than 0.1 (Šimundić, 2009).

Another global metric for the comparison of diagnostic tests is the diagnostic odds ratio (DOR). It represents the ratio between LR+ and LR-(97), or the ratio of the probability of a positive test result if the sample has the disease to the likelihood of a positive result if the sample does not have the disease. DOR can range from zero to infinity, and a test is only useful with values larger than 1.0 (Glas et al., 2003). The last metrics used in this work are commonly used to evaluate machine learning classification results. The F1-score, also known as F-measure, ranges from zero to one and is the harmonic mean of the precision and recall (Tharwat, 2020). The area under the receiver operating characteristic (AUROC) describes the model’s ability to discriminate between positive and negative examples measuring the trade-off between the true positive rate and the false positive rate across different thresholds.

Machine learning approaches

Among several ML applications in real-world situations, classification tasks stand up as one of the most relevant applications, ranging from classification of types of plants and animals to the identification of different diseases prognoses, such as cancer (Feltes et al., 2019; Feltes, Poloni & Dorn, 2021; Feltes et al., 2020; Grisci, Feltes & Dorn, 2019), H1N1 Flu (Chaurasia & Dixit, 2021), Dengue (Zhao et al., 2020), and COVID-19 (Table 5). The use of these algorithms in the context of hemogram data from COVID-19 patients is summarized in Table 5.

Table 5 Studies that use ML algorithms on COVID-19 hemogram data (in alphabetical order by the surname of the first author).

Source	Data	Algorithms	
AlJame et al. (2020)	CBC, Albert Einstein Hospital, Brazil	XGBoost	
Alves et al. (2021)	CBC, Albert Einstein Hospital, Brazil	Random Forest, Decision Tree, Criteria Graphs	
Assaf et al. (2020)	Clinical and CBC profile, Sheba Medical Center, Israel	MLP, Random Forest, Decision Tree	
Avila et al. (2020)	CBC, Albert Einstein Hospital, Brazil	Nave-Bayes	
Banerjee et al. (2020)	CBC, Albert Einstein Hospital, Brazil	MLP, Random Forest, Logistic Regression	
Bao et al. (2020)	CBC, Wuhan Union Hosp; Kunshan People’s Hosp, China	Random Forest, SVM	
Bhandari et al. (2020)	Clinical and CBC profile of (non) survivors, India	Logistic Regression	
Brinati et al. (2020b)	CBC, San Raffaele Hospital, Italy	Random Forest, Nave-Bayes, Logistic Regression, SVM, kNN	
Cabitza et al. (2020)	CBC, San Raffaele Hospital, Italy	Random Forest, Nave-Bayes, Logistic Regression, SVM, kNN	
Delafiori et al. (2021)	Mass spectrometry, COVID-19, plasma samples, Brazil	Tree Boosting, Random Forest	
de Freitas Barbosa et al. (2020)	CBC, Albert Einstein Hospital, Brazil	MLP, SVM, Random Forest, Nave-Bayes	
Joshi et al. (2020)	CBC of patients from USA and South Korea	Logistic Regression	
Silveira (2020)	CBC, Albert Einstein Hospital, Brazil	XGBoost	
Shaban et al. (2020)	CBC, San Raffaele Hospital, Italy	Fuzzy inference engine, Deep Neural Network	
Soares et al. (2020)	CBC, Albert Einstein Hospital, Brazil	SVM, SMOTEBoost, kNN	
Yan et al. (2020)	Laboratory test results and mortality outcome, Wuhan	XGBoost	
Zhou, Chen & Lei (2020)	CBC, Tongji Hospital, China	Logistic Regression	

The number of features and characteristics of different datasets might be a barrier for distinctive classification learning techniques. Furthermore, it is of extreme importance a better understanding and characterization of the strengths and drawbacks of each classification technique used (Kotsiantis, Zaharakis & Pintelas, 2006). The following classifiers’ choice was based on their use as listed in Table 5, as they are the most likely to be used in experiments with COVID-19 data.

Naïve Bayes

One of the first ML classification techniques is based on the Bayes theorem (Eq. (3)). The Naïve Bayes classification technique is a probabilistic classifier that calculates a set of probabilities by counting the frequency and combinations of values in the dataset. The Naïve Bayes classifier has the assumption that all attributes are conditionally independent, given the target value (Huang & Li, 2011).

(3) P(A|B)=P(A)P(B|A)P(B)

where P(A) is the probability of the occurrence of event A, P(B) is the probability of occurrence of event B, and P(A|B) is the probability of occurrence of event A when B also occurs. Likewise, P(B|A) is the probability of event B when A also occurs.

In imbalanced datasets, the Naïve Bayes classification algorithm biases the major class results in the dataset, as it happens with most of the classification algorithms. To handle the imbalanced data set in biomedical applications, the work of Min et al. (2009) evaluated different sampling techniques with the NB classification. The used sampling techniques did not show a significant difference in comparison with the imbalanced data set.

Support vector machines

Support Vector Machine (SVM) (Cortes & Vapnik, 1995) is a classical supervised learning method for classification that works by finding the hyperplane (being just a line in 2D or a plane in 3D) capable of splitting data points into different classes. The “learning” consists of finding a separating hyperplane that maximizes the distance between itself and the closest data points from each class, called the support vectors. In the cases where the data is not linearly separable, kernels are used to transform the data by mapping it to higher dimensions where a separating hyperplane can be found (Harrington, 2012). SVM usually performs well on new datasets without the need for modifications. It is also not computationally expensive, has low generalization errors, and is interpretative in the case of the data’s low dimensionality. However, it is sensitive to kernel choice and parameter tuning and can only perform binary classification without algorithmic extensions (Harrington, 2012).

Although SVM achieves impressive results in balanced datasets, when an imbalanced dataset is used, the rating performance degrades as with other methods. In Batuwita & Palade (2013), it was identified that when SVM is used with imbalanced datasets, the hyperplane is tilted to the majority class. This bias can cause the formation of more false-negative predictions, a significant problem for medical data. To minimize this problem and reduce the total number of misclassifications in SVM learning, the separating hyperplane can be shifted (or tilted) to the minority class (Batuwita & Palade, 2013). However, in our previous study, we noticed that for curated microarray gene expression analyzes, even in imbalanced datasets, SVM generally outperformed the other classifiers (Feltes et al., 2019). Similar results were highlighted in other reviews (Ang et al., 2016).

K-nearest neighbors

The nearest neighbor algorithm is based on the principle that instances from a dataset are close to each other regarding similar properties (Kotsiantis, Zaharakis & Pintelas, 2006). In this way, when unclassified data appears, it will receive the label accordingly to its nearest neighbors. The extension of the algorithm, known as k-Nearest Neighbors (kNN), considers a parameter k, defining the number of neighbors to be considered. The class’s determination is straightforward, where the unclassified data receives the most frequent label of its neighbors. To determine the k nearest neighbors, the algorithm considers a distance metric. In our case, the Euclidean Distance (Eq. (4)) is used:

(4) D(x,y)=∑i=1n|xi−yi|2

where x and y are two instances with n comparable characteristics. Although the kNN algorithm is a versatile technique for classification tasks, it has some drawbacks, such as determining a secure way of choosing the k parameter, being sensitive to the similarity (distance) function used (Kotsiantis, Zaharakis & Pintelas, 2006), and a large amount of storage for large datasets (Harrington, 2012). As the kNN considers the most frequent class of its nearest neighbors, it is intuitive to conclude that for imbalanced datasets, the method will bias the results towards the majority class in the training dataset (Kadir et al., 2020).

For biological datasets, kNN is particularly useful for data from non-characterized organisms, where there is little-to-non previous information to identify molecules and their respective bioprocesses correctly. Thus, this “guilty by association” approach becomes necessary. This logic can be extrapolated to all types of biological datasets that possess such characteristics.

Decision trees

Decision trees are one of the most used techniques for classification tasks (Harrington, 2012), although they can also be used for regression. Decision trees classifies data accordingly to their features, where each node represents a feature, and each branch represents the value that the node can assume (Kotsiantis, Zaharakis & Pintelas, 2006). A binary tree needs to be built based on the feature that better divides the data as a root node to classify data. New subsets are created in an incremental process until all data can be categorized (Harrington, 2012). The first limitation of this technique is the complexity of constructing a binary tree (considered an NP-Complete problem). Different heuristics were already proposed to handle this, such as the CART algorithm (Breiman et al., 1984). Another important fact is that decision trees are more susceptible to overfitting (Harrington, 2012), requiring the usage of a pruning strategy.

Since defining features for splitting the decision tree is directly related to the training model performance, knowing how to treat the challenges imposed by imbalanced datasets is essential to improve the model performance, avoiding bias towards the majority class. The effect of imbalanced datasets in decision trees could be observed in Cieslak & Chawla (2008). The results attested that decision tree learning models could reach better performance when a sampling method for imbalanced data is applied.

Random forest

Random Forests are an ensemble learning approach that uses multiple non-pruned decision trees for classification and regression tasks. To generate a random forest classifier, each decision tree is created from a subset of the data’s features. After many trees are generated, each tree votes for the class of the new instance (Breiman, 2001). As random forest creates each tree based on a bootstrap sample of the data, the minority class might not be represented in these samples, resulting in trees with poor performance and biased towards the majority class (Chen, Liaw & Breiamn, 2004). Methods to handle the high-imbalanced data were compared by Chen, Liaw & Breiamn (2004), including incorporating class level weights, making the learning models cost-sensitive, and reducing the amount of the majority class data for a more balanced data set. In all cases, the overall performance increased.

XGBoost

The XGBoost framework was created by Chen & Guestrin (2016), and is used on decision tree ensemble methods, following the concept of learning from previous errors. More specifically, the XGBoost uses the gradient of the loss function in the existing model for pseudo-residual calculation between the predicted and real label. Moreover, it extends the gradient boosting algorithm into a parallel approach, achieving faster training models than other learning techniques to maintain accuracy.

The gradient boosting performance in imbalanced data sets can be found in Brown & Mues (2012), where it outperforms other classifiers such as SVM, decision trees, and kNN in credit scoring analysis. The eXtreme Gradient Boost was also applied to credit risk assignment with imbalanced datasets in Chang, Chang & Wu (2018), achieving better results than its competitors.

Logistic regression

Logistic regression is a supervised classification algorithm that builds a regression model to predict the class of a given data based on a Sigmoid function (Eq. (5)). As occurs in linear models, in logistic regression, learning models compute a weighted sum of the input features with a bias (Géron, 2017). Once the logistic model estimated the probability of p of a given data label, the label with p ≥ 50% will be assigned to the binary classification data.

(5) g(z)=11+e−z

Multilayer perceptron

A multilayer perceptron is a fully connected neural network with at least three layers of neurons: one input layer, one hidden layer, and an output layer. The basic unit of a neural network is a neuron that is represented as nodes in the neural network, and have an activation function, generally, a Sigmoid function (Eq. (5)), which is activated accordingly to the sum of the arriving weighted signals from previous layers.

For classification tasks, each output neuron represents a class, and the value reported by the i-th output neuron is the amount of evidence in supported i-th class (Kubat, 2017), i.e., if an MLP has two output neurons—meaning that there are two classes—the output evidence could be (0.2, 0.8), resulting to the classification of the class supported by the highest value, in this case, 0.8. Based on the learning model prediction’s mean square error, each connection assigned weights are adjusted based on the backpropagation learning algorithm (Kubat, 2017). Although the MLPs have shown impressive results in many real-world applications, some drawbacks must be highlighted. The first one is the determination of the number of hidden layers. An underestimation of the neurons number can cause a poor classification capability, while the excess of them can lead to an overfitting scenario, compromising the model generalization. Another concern is related to the computational cost of the backpropagation, where the process of minimizing the MSE takes long runs of simulations and training. Furthermore, one of the major characteristics is that MLPs are black-box methods, making it hard to understand the reason for their output (Kotsiantis, Zaharakis & Pintelas, 2006).

Regarding the capabilities of MLPs in biased data, an empirical study is provided by Khoshgoftaar, Van Hulse & Napolitano (2010), showing that MLP can achieve satisfactory results in noisy and imbalanced datasets even without sampling techniques for balancing the datasets. The analysis provided by the authors showed that the difference between the MLP with and without sampling was minimal.

Techniques to handle imbalanced data

As introduced before, the COVID-19 CBC data is highly imbalanced. In a binary classification problem, class imbalance occurs when one class, the minority group, contains significantly fewer samples than the other class, the majority group. In such a situation, most classifiers are biased towards the larger classes and have meager classification rates in the smaller classes. It is also possible that the classifier considers everything as the largest class and ignores the smaller class. This problem is faced not only in the binary class data but also in the multi-class data (Tomašev & Mladenić, 2013).

A significant number of techniques have been proposed in the last decade to handle the imbalanced data problem. In general, we can classify these different approaches as sampling methods (pre-processing) and cost-sensitive learning (Haixiang et al., 2017). In cost-sensitive learning models, the minority class misclassification has a higher relevance (cost) than the majority class instance misclassification. Although this can be a practical approach for imbalanced datasets, it can be challenging to set values for the needed matrix cost (Haixiang et al., 2017).

The use of sampling techniques is more accessible than cost-sensitive learning, requiring no specific information about the classification problem. For these approaches, a new dataset is created to balance the classes, giving the classifiers a better opportunity to distinguish the decision boundary between them (He & Ma, 2013). In this work, the following sampling techniques are used, chosen due to their prominence in the literature: Random Over-Sampling (ROS), Random Under-Sampling (RUS), Synthetic Minority Over-sampling TEchnique (SMOTE), Synthetic Minority Over-sampling Technique with Tomek Link (SMOTETomek), and Adaptive Synthetic Sampling (A-DASYN). All of them are briefly described in this section. A t-SNE visualization of each sampling technique’s effect for the three datasets used can be seen in Fig. 4.

Figure 4 Visualization of the negative (purple) and positive (green) samples from the Albert Einstein Hospital (AE), Fleury Laboratory (FLEURY) and Hospital Sirio Libanês (HSL) using t-SNE for all the different sampling schemes.

Random sampling

In classification tasks that use imbalanced datasets, sampling techniques became standard approaches for reducing the difference between the majority and minority classes. Among different methods, the most simpler ones are the RUS and ROS. In both cases, the training dataset is adjusted to create a new dataset with a more equanimous class distribution (He & Ma, 2013).

For the under-sampling approach, most class instances are discarded until a more balanced data distribution is reached. This data dumping process is done randomly. Considering a dataset with 100 minority class instances and 1,000 majority class instances, a total of 900 majority class instances would be randomly removed in the RUS technique. At the end of the process, the dataset will be balanced with 200 instances. The majority class will be represented with 100 instances, while the minority will also have 100.

In contrast, the random over-sampling technique duplicates minority class data to achieve better data distribution. Using the same example given before, with 100 instances of the minority class and 1,000 majority class instances, each data instance from the minority class would be replicated ten times until both classes have 1,000 instances. This approach increases the number of instances in the dataset, leading us to 2,000 instances in the modified dataset.

However, some drawbacks must be explained. In RUS, the data dumping process can discard a considerable number of data, making the learning process harder and resulting in poor classification performance. On the other hand, for ROS, the instances are duplicated, which might cause the learning model overfitting, inducing the model to a lousy generalization capacity and, again, leading to lower classification performance (He & Ma, 2013).

Synthetic minority over-sampling technique (SMOTE)

To overcome the problem of generalization resulting from the random over-sampling technique (Chawla et al., 2002) created a method to generate synthetic data in the dataset. This technique is known as SMOTE. To balance the minority class in the dataset, SMOTE first selects a minority class data instance Ma randomly. Then, the k nearest neighbors of Ma, regarding the minority class, are identified. A second data instance Mb is then selected from the k nearest neighbors set. In this way, Ma and Mb are connected, forming a line segment in the feature space. The new synthetic data is then generated as a convex combination between Ma and Mb. This procedure occurs until the dataset is balanced between the minority and majority classes. Because of the effectiveness of SMOTE, different extensions of this over-sampling technique were created.

As SMOTE uses the interpolation of two instances to create the synthetic data, if the minority class is sparse, the newly generated data can result in a class mixture, which makes the learning task harder (Branco, Torgo & Ribeiro, 2016). Because SMOTE became an effective over-sampling technique and still has some drawbacks, different variations of the method were proposed by different authors. A full review of these different types can be found in Branco, Torgo & Ribeiro (2016) and He & Ma (2013).

Synthetic minority over-sampling technique with Tomek link

Although the SMOTE technique achieved better results than random sampling methods, data sparseness can be a problem, particularly in datasets containing a significant outlier occurrence. In many datasets, it is possible to identify that different data classes might invade each class space. When considering a decision tree as a classifier with this mixed dataset, the classifier might create several specialized branches to distinguish the data class (Batista, Bazzan & Monard, 2003). This behavior might create an over-fitted model with poor generalization.

In light of this fact, the SMOTE technique was extended considering Tomek links (Tomek, 1976) by Batista, Bazzan & Monard (2003) for balancing data and creating more well-separated class instances. In this approach, every data instance that forms a Tomek link is discarded, both from minority and majority classes. A Tomek link can be defined as follows: given two samples with different classes SA and SB, and a distance d(SA, SB), this pair (SA, SB) is a Tomek link if there is not a case SC that d(SA, SC) < d(SA, SB) and d(SB, SC) < d(SB, SA). In this way, noisy data is removed from the dataset, improving the capability of class identification.

In the SMOTE technique, the new synthetic samples are equally created for each minority class data point. However, this might not be an optimized way to produce synthetic data since it can concentrate most of the data points in a small portion of the feature space.

Adaptive synthetic sampling

Using the adaptive synthetic sampling algorithm, ADASYN (He et al., 2008), a density estimation metric is used as a criterion to decide the number of synthetic samples for each minority class example. With this, it is possible to balance the minority and majority classes and create synthetic data where the samples are difficult to learn. The synthetic data generation occurs as follows: the first step is to calculate the number of new samples needed to create a balanced dataset. After that, the density estimation is obtained by the k-nearest neighbors for each minority class sample (Eq. (6)) and normalization (Eq. (7)). Then the number of needed samples for each data point is calculated (Eq. (8)), and new synthetic data is created.

(6) ri=ΔiK,i=1,...,ms

(7) r^i=ri∑i=1msri

(8) gi=r^i×G

where ms is the set of instances representing the minority classes, Δi the number of examples in the K nearest neighbors belonging to the majority class, gi defines the number of synthetic samples for each data point, and G is the number of synthetic data samples that need to be generated to achieve the balance between the classes.

Experiments and results

To evaluate the impact of the data imbalance on the Brazilian CBC datasets, we have applied the sampling techniques described in “Techniques to Handle Imbalanced Data”. They are discussed in three different aspects. The first one is the comparison between classification methods without resampling. In this way, we can compare how each classifier deals with the imbalance. The second aspect is related to the sampling methods of efficiency compared to the original datasets.

Each classification model was trained with the same training set (70% of samples) and was tested to the same test set (30% of samples). The features were normalized using the z-score. Evaluation metrics were generated by 31 runs considering random data distribution in each partition. The proposed approach was implemented in Python 3 using Scikit-Learn (Pedregosa et al., 2011) as a backend. The COVID-19 classes were defined using RT-PCR results from the datasets. Sampling techniques were applied only on the training set. Hyperparameters were optimized using the Randomized Parameter Optimization approach available in scikit-learn and the values in Table 6. The aim of optimizing the hyperparameters is to find a model that returns the best and most accurate performance obtained on a validation set. Figure 1 schematizes the methodological steps used in this work.

Table 6 Hyperparameter ranges used in our analyses.

Classifier	Parameters	
Naive Bayes	–	–	
Support Vector Machines	kernel:	rbf; linear	
	gama:	0.0001–0.001	
	c:	1–1,000	
Random Forest	n-estimators:	50; 100; 200	
	criterion:	gini; entropy	
	max_depth:	3–10	
	min_samples_split:	0.1–0.9	
XgBoost	n-estimators:	50; 100; 200	
	max_depth:	3–10	
	learning_rate:	0.0001–0.01	
Decision Tree	criterion:	gini; entropy	
	max_depth:	3–10	
	min_samples_split:	0.1–1.0	
K-Nearest Neighbors	n_neighbors:	3; 5; 7; 10; 15; 50	
	weights:	uniform; distance	
Logistic Regression	–	–	
Multi Layer Perceptron	activation:	logistic; tanh; relu	
	solver:	sgd; adam	
	alpha:	0.0001; 0.001; 0.01	
	learning_rate_init:	0.0001; 0.001; 0.01	
	early_stopping:	True; False	
	batch_size:	16; 64; 128	
	hidden_layer_sizes:	(10, 10, 2); (5, 10, 5); (10); (10, 20, 5); (10, 10); (100); (30, 10)	

Different results were obtained for each classification method with an imbalanced dataset, as can be seen in Figs. 4–61 . In terms of F1 Score all classification models achieved values ranging around 0.5 to 0.65 for the Albert Einstein and Fleury datasets. Although the F1 Score is widely used to evaluate classification tasks, it must be carefully analyzed in our case since the misclassification has more impact, especially in false-negative cases, making it necessary to observe other indexes. When the sensitivity index is considered, it draws attention to the disparity between the NB classification model and the others.

Figure 5 Average test results from 31 independent runs for several classifiers and sampling schemes trained on the Albert Einstein Hospital data. Black lines represent the standard deviation, while the white circle represents the median. (A) Sensitivity; (B) Specificity; (C) LR+; (D) LR−; (E) DOR; (F) F1 Score; (G) ROC-AUC Score.

Figure 6 Average test from 31 independent runs for several classifiers and sampling schemes trained on the Fleury Group data. Black lines represent the standard deviation, while the white circle represents the median. (A) Sensitivity; (B) Specificity; (C) LR+; (D) LR−; (E) DOR; (F) F1 Score; (G) ROC-AUC Score.

The NB model achieved a sensitivity of around 0.77 for the Albert Einstein dataset (Fig. 4A) and around 0.72 for the Fleury dataset (Fig. 6A). Hence, it is possible to consider the NB as the classification model to better detect the true positive cases (minority class) in these data sets. However, when considering the specificity (Figs. 5B and 6B), it is notable that NB achieved the worst performance overall. A possible explanation for this disparity is that NB classifies most of the data as positive for possible SARS-CoV-2 infection. This hypothesis is then confirmed when we analyze the other two indexes (DOR and LR+), showing NB bias to the minority class. When considering other classification models regarding sensitivity and specificity, the LR, RF, and SVM achieved better results, ranging from 0.55 to 0.59 for sensitivity and 0.89 to 0.93 for specificity. This better balance between sensitivity and sensibility is mirrored in the F1 Score, where RF, SVM, and LR achieved better performance than other methods (and comparable to NB) while achieving better DOR and LR+.

When considering four key indexes (F1 Score, ROC-AUC Score, Sensitivity, and Specificity), we can observe that the sampling techniques improved the learning models regarding the classification of positive cases of SARS-CoV-2 from the Albert Einstein dataset in comparison with the original data, except for NB. Thus, reducing the bias to the majority class observed in the original data set, especially when considering the specificity (the proportion of correctly classified negative samples among all negative samples). For Albert Einstein and Fleury datasets, sampling techniques improve the sensitivity and lower all classifiers’ specificity. For the HSL dataset, we see the opposite; resampling decreases the sensitivity and improves the specificity. This happens because while for Albert Einstein and Fleury, the majority class is negative, the majority class is positive for HSL.

Furthermore, with sampling techniques, the DOR was improved in the Albert Einstein dataset. With Fleury data, the learning models with sampling did not achieve tangible DOR results. A possible explanation of this outcome can be related to the data sparseness, an ordinary circumstance observed in medical or clinical data. This is further corroborated by the data visualization using t-SNE in Fig. 4. Moreover, the number of samples used with the Fleury dataset could be determinant for the poor performance. Nevertheless, overall, no sampling technique appears to be a clear winner, especially considering the standard deviation. The performance of each sampling technique is conditioned by the data, metric, and classifier at hand.

Regarding the decrease in LR+ when the Albert Einstein or Fleury data is balanced, LR+ represents the probability of samples classified as positive being truly positive. The difference of LR+ values in the original datasets compared to the resampled data is due to the classifier trained on the original data labeling most samples as negative, even when facing a positive sample. Thus, it is important to note that when the data is balanced, the bias towards the negative class diminishes, and the model has more instances being classified as (true or false) positives.

None of the combinations of classifiers and sampling methods achieved satisfactory results for the Sírio-Libanês Hospital dataset (Fig. 7). The sensitivity of all options was close to one, and the specificity was close to zero, indicating that almost all samples are being predicted as the majority class (in this case, the positive). This was expected due to the large imbalance of this dataset, and even the sampling methods, although able to narrow the gap, were not enough to achieve satisfactory results. Due to these poor results, the other metrics are non-satisfactory, and their results can be misleading. For instance, if one were only to check the F1 Score, the classification results would seem satisfactory. As listed in Table 1 and illustrated in Fig. 7, this dataset had the largest imbalance, with over forty times more positive than negative samples. Moreover, the total number of available samples was the smallest among the three datasets. The results suggest that using standard ML classifiers is not useful for such drastic cases even when sampling techniques are applied, and researchers should be cautious when dealing with similar datasets (low sample quantity and high imbalanced data).

Figure 7 Average test results from 31 independent runs for several classifiers and sampling schemes trained on the Sírio-Libanês Hospital. Black lines represent the standard deviation, while the white circle represents the median. (A) Sensitivity; (B) Specificity; (C) LR+; (D) LR−; (E) DOR; (F) F1 Score; (G) ROC-AUC Score.

The results obtained in our simulations showed that ML classification techniques could be applied as an assistance tool for COVID-19 diagnosis in datasets with a large enough number of samples and moderate levels of imbalance (less than 50%), even though some of them achieved poor performance or biased results. It is essential to notice that the NB algorithm reached better classification when targeting the positive cases for SARS-CoV-2. However, it skews the classification in favor of the minority class. Hence, we believe that SVM, LR, and RF approaches are more suitable to the problem.

Future research can be conducted with these limitations in mind, building ensemble learning models with RF, SVM, and LR, and different approaches to handle the imbalanced data sets, such as the use of cost-sensitive methods. It is also important to note that some of these classifiers, such as MLP, cannot be considered easily interpretable. This presents a challenge for their use of medical data, in which one should be able to explain their decisions. Both issues could be tackled in the future using feature selection (Ang et al., 2016; Grisci, Feltes & Dorn, 2019) or algorithms for explainable artificial intelligence (Yang et al., 2018; Montavon, Samek & Müller, 2018; Arga, 2020). The method of relevance aggregation, for instance, can be used to extract which features from tabular data were more relevant for the decision making of neural networks and was shown to work on biological data (Grisci, Krause & Dorn, 2021). Feature selection algorithms can also be used to spare computational resources by training smaller models and to improve the performance of models by removing useless features.

Conclusions

The COVID-19 pandemic has significantly impacted countries that cannot test their population and develop strategies to manage the crisis and those with substantial financial limitations. Artificial intelligence and ML play a crucial role in better understanding and addressing the COVID-19 emergency and devising low-cost alternatives to aid decision making in the medical field. In this sense, ML techniques are being applied to analyze different data sources seeking to identify and prioritize patients tested by RT-PCR.

Some features that appear to be the most representative of the three analyzed datasets are basophils and eosinophils, which are among the expected results. The work of Banerjee et al. (2020) showed that patients displayed a significant decrease in basophils, as well as eosinophils, something also discussed in other works (Bayat et al., 2020).

Having imbalanced data is common, but it is especially prevalent when working with biological datasets, and especially with disease data, where we usually have more healthy control samples than disease cases, and an inherent issue in acquiring clinical data. This work reviews the leading ML methods used to analyze CBC data from Brazilian patients with or without COVID-19 by different sampling and classification methods.

Our results show the feasibility of using these techniques and CBC data as a low-cost and widely accessible way to screen patients suspected of being infected by COVID-19. Overall, RF, LR, and SVM achieved the best general results, but each classifier’s efficacy will depend on the evaluated data and metrics. Regarding sampling techniques, they can alleviate the bias towards the majority class and improve the general classification, but no single method was a clear winner. This shows that the data should be evaluated on a case-by-case scenario. More importantly, our data point out that researchers should never rely on the results of a single metric when analyzing clinical data since they show fluctuations, depending on the classifier and sampling method.

However, the application of ML classifiers, with or without sampling methods, is not enough in the presence of datasets with few samples available and large class imbalance. For such cases, that more often than not are faced in the clinical practice, ML is not yet advised. If the data is clearly biased, like the HSL data, the dataset should be discarded. Even for adequate datasets and algorithms, the selection of proper metrics is fundamental. Sometimes, the values can camouflage biases in the results and poor performance, like the NB classifier’s case. Our recommendation is to inspect several and distinct metrics together to see the greater picture.

Additional Information and Declarations

Competing Interests

Author Contributions

Data Availability

1 The statistical comparison between the algorithms (Dunn’s Multiple Comparison Test with Bonferroni correction) is available in the GitHub repository: https://github.com/sbcblab/sampling-covid.

The authors declare that they have no competing interests.

Marcio Dorn conceived and designed the experiments, performed the experiments, analyzed the data, performed the computation work, prepared figures and/or tables, authored or reviewed drafts of the paper, and approved the final draft.

Bruno Iochins Grisci conceived and designed the experiments, analyzed the data, performed the computation work, prepared figures and/or tables, authored or reviewed drafts of the paper, and approved the final draft.

Pedro Henrique Narloch conceived and designed the experiments, analyzed the data, performed the computation work, prepared figures and/or tables, authored or reviewed drafts of the paper, and approved the final draft.

Bruno César Feltes analyzed the data, prepared figures and/or tables, authored or reviewed drafts of the paper, and approved the final draft.

Eduardo Avila conceived and designed the experiments, analyzed the data, authored or reviewed drafts of the paper, and approved the final draft.

Alessandro Kahmann analyzed the data, authored or reviewed drafts of the paper, and approved the final draft.

Clarice Sampaio Alho analyzed the data, authored or reviewed drafts of the paper, and approved the final draft.

The following information was supplied regarding data availability:

The source code used for the experiments are available at GitHub:

https://github.com/sbcblab/sampling-covid.

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
