# Peer review of "Comparison of machine learning techniques to handle imbalanced COVID-19 CBC datasets"

_PeerJ Computer Science, doi:10.7717/peerj-cs.670_

## Round 0.1 · original submission · Minor Revisions

Please take a look at the reviewers' suggestions.

Reviewer 1 ·

Basic reporting

The authors follow all the suggested guidances: clear and unambiguous, professional English used throughout; literature references, sufficient field background/context provided; professional article structure, figures, tables; raw data shared; self-contained with relevant results to hypotheses; and formal results should include clear definitions of all terms and theorems, and detailed proofs.

Experimental design

The authors also follow the guidance of this area: original primary research within Aims and Scope of the journal; Research question well defined, relevant & meaningful; rigorous investigation performed to a high technical & ethical standard; and methods described with sufficient detail & information to replicate.

Validity of the findings

The authors again follow the guidelines: all underlying data have been provided; and the conclusions are well stated, linked to original research question & limited to supporting results.

Additional comments

This is a study of a public database of covid-19 in Brazil with the objective of comparing various techniques of data rebalancing. This is a strategy routinely carried out in most health studies, since the imbalance between classes is frequent in this area. The article does not bring new insights to the area, but it is technically correct and provides a good tutorial on recent practical applications of machine learning algorithms.

The major problem is the quality of the database provided. Although the data comes from renowned health institutions in Brazil, there have been several reports of inconsistencies in this database, which is the reason why many ML groups have avoided using them. For example, the positive class for covid-19 is not expected to be 40 times more prevalent than the negative (as in the case of HSL), even if only symptomatic cases are tested.

The authors address this issue by analyzing the bivariate distribution of variables according to the two classes. I believe that in this case it would be more appropriate to use statistical analysis to compare whether the variables have similar distributions for the three centers, regardless of the outcome.

Another important issue is that the authors present the results of several predictive metrics, but not the one most used in machine learning models (AUROC). I believe this metric should be used as the main result of the analysis.

·

Basic reporting

I strongly suggest a review in English for the article. Some adjustments are necessary to avoid long and redundant sentences. These adjustments will make the article more straightforward and easier to read.

The database used has information on a large number of patient samples from a highly mixed population. The authors can discuss more deeply the influence of this kind of variability in obtaining predictive models. Likewise, they can describe the strengths and weaknesses of this variability in obtaining predictive models with good performance.

The manuscript is not in exactly the same format suggested by the Journal. There is no discussion session. The discussion took place in the results session.

Experimental design

Thinking about different reference values between men and women, as well, the age impact on the CBC values, did you evaluate the impact of adding sex information as a feature?

In addition to eliminating samples that did not have all CBC records, and using the complete blood count of the PCR date, was any other manipulation of the data done by the authors? It was not clear in the text whether the authors did extra preprocessing of the datasets.

In line 137 the authors described the features and scale. Were the scales the same among the 3 datasets? Did the authors perform any scale process in these datasets?

The time period during which these data sets were collected isn't mentioned, were the records collected over which period of time?

How the authors defined the positive diagnosis for SARS Cov 2 is not described in the manuscript. Was the COVID-19 class defined using RT-PCR result?

Is the y column in the datasets the SARS Cov 2 classification? The authors can describe the class definition with more details.

Validity of the findings

The authors did not present any statistical test showing whether there is or not a significant difference between the assessments obtained with each method and/or algorithm. I recommend the addiction of the statistical test results. This information will help to better guide the discussion. Furthermore, in this way the evaluation does not appear to be only qualitative or even arbitrary.

Starting in line 214 the authors discuss the possible influence of seasonal fluctuations and, sometimes, extreme circadian regulations in the datasets variability. However, the datasets were obtained from a population from the same country. Therefore, it would be interesting to make clear the characteristics of this population. Subsequently, describe the possible influence of seasonal fluctuations and extreme circadian regulations, mainly on datasets obtained from different populations.

Line 548:“Both issues could be tackled in the future using feature selection
Line 549: (3; 48) or algorithms for explainable artificial intelligence (100; 76; 5).” The authors discuss the need for feature selection mainly for the interpretability of models obtained with Multi-Layer Perceptron (MLP). However, at no time do they discuss the possibility of applying feature selection methods to improve the predictive performance of the model. Or even to simplify the model, and consequently reduce the use of computational resources.

Additional comments

Main comment:
The study is interesting, mainly, due to the need for more accessible alternatives for prioritizing patients to perform the RT-PCR test. The CBC is an interesting biological information base. In addition, the database used has information on a large number of patient samples from a highly mixed population. What makes the datasets valuable. However, the authors found many differences between the classification results obtained with the 3 datasets. In this way, they could have explored the possibility of obtaining predictive models from the integration of the 3 datasets. I understand the limitation and the difficulty of this approach, especially when dealing with such diverse distributions, but it could be a good complement to the work. Show the impact of generalization and the increase in the number of samples on the performance of predictive models. In order to assist other researchers in decision making. They can decide to use only a single dataset obtained from the same institution, or to use more datasets to define their predictive covid-19 models.

Comments:
The time period during which these data sets were collected isn't mentioned, were the records collected over which period of time?

In the line 65 and 66 the authors talked about some works, but you mentioned only one.
“and some works suggest routine blood tests as a potential diagnostic tool for 66 COVID-19 (40).“ The authors can fix the phrase or add more references to the sentence.

Line 73: “Recently, artificial intelligence techniques, especially Machine Learning (ML), have been employed to analyze CBC data and assist in screening of patients with suspected COVID-19 infection (99; 101; 47; 1;7; 19; 60; 96)”. The authors cite in the text several studies using CBC data to COVID-19 patients screening, but they did not describe the findings of these works. Besides, they also do not discuss how the findings of the other authors corroborate their findings in this work.

Figure 4: I suggest that the authors arrange the distribution of the t-SNE visualization. Place each sampling method in a line and each dataset in a column. This will make it easier to compare sampling methods between datasets. So, instead of having 5 rows and 4 columns, the Figure would have 6 rows and 3 columns.

Supplementary figures comparing the same sampling method and the same algorithm for the different datasets could be interesting. It would make visualization easier when comparing datasets and not just methods.

In the same sense, a stacked Bar Chart showing the number of samples and the proportion of positive and negative COVID-19 records would be interesting. This figure would facilitate the reader's understanding of the number of samples and their distribution among the datasets.

Minor changes:

In the datasets available in: https://github.com/sbcblab/sampling-covid/blob/main/AE/DATA The features sexo and idade, are written in portuguese. Please, translate to english for keeping all features in the same pattern.
Line 103: typo: “TEchnique”
Line 116: “more than ≈ 13 million cases” The ≈ is not necessary and the numbers are more than 14 millions in this moment.
Line 138: typo: “pla-telets
Line 141: typo: “red blo-od”
Line 142: typo: “% Eosino-phils”
Line: 143 “Patients with incomplete (missing data) or no data available for the above parameters were not included in the present analysis. For patients with more than a single test result available, a unique hemogram test was used, with the selection based on the blood test date - same-day results to the PCR-test collection date was adopted as a reference, or the day closest to this date.” These sentences must be rewritten to facilitate the reader's understanding.
Line 198: typo: “FLEU-RY”
Line 261: “the identification of different diseases prognoses, such as cancer (39; 48) and COVID-19 (7)”. It would be interesting to cite more examples.
Line 300: ”Similar results were highlighted in other reviews (3)”. It would be better to cite more reviews or write in the singular.
Line 330: “Since defining features for splitting the decision tree is directly related to the training model performance, dealing with imbalanced datasets is essential to improve the model performance, avoiding bias towards the majority class.” The meaning of the phrase at first reading seems controversial. It makes us believe that they want to use the unbalanced data, and not deal with the date in order to correct this imbalance. A slight adjustment in English would avoid this mistake on the part of the reader.
401 typo: “Synthetic Minority Over-sampling Technique (SMO-TE)”
Line 559: It would be better to remove the citation in the conclusion topic
Line 561: It would be better to remove the citation in the conclusion topic
F1 Score sometimes is written with a different font and sometimes not.

Reviewer 3 ·

Basic reporting

The manuscript is clear, very well written and organized. It explores the pros and cons of different machine learning techniques to handle imbalanced biological datasets (in this case, focused on blood sampling related to COVID19).
The literature references are adequate, and the background information is detailed enough for non-experts to understand the work presented. All figures are clear and informative, summarizing the results and the overall strategy employed. All definitions necessary are presented in a clear fashion.

Experimental design

The research question is clearly stated, and well grounded by the literature cited.
It fills a gap in the knowledge regarding machine learning applied to clinical data and diagnostics. All methods are clearly stated.

Validity of the findings

The results are sound and the conclusions are clearly stated. All data required to support the conclusions is presented in the article.

Additional comments

This is a very well written article, dealing with a critical step in the use of machine learning for diagnostic purposes.

I see only minor points to be corrected:
l. 3 (title): define CBC
l. 29: change "to CBC" to "for CBC"
l. 31: change "evaluated" to "evaluate"
l. 61: change "most optimal" to "optimal"
l. 68: remove nominal reference
l. 91: change "applications of different" to "applications on different"
l. 142: remove dash from "Eosino-phils"
l. 161: change "founds" to "funds"
l. 190: change "Complementary" to "Complementarily"
l. 198: remove dash from "FLEU-RY"
l. 207: change "too" to "as well"
l. 212: remove nominal reference
l. 214: remove nominal reference
l. 234: change "allows a more" to "allows for a more"
l. 397: remove dash from "ab-out"
l. 428: remove "issue"
l. 483: change "the best and accurate" to "the best and most accurate"
l. 496, 505, 542: change "Sars-Cov-2" to "SARS-CoV-2"
l. 574: change "point-out" to "point out"
l. 577: change "carefulness to be taken" to "carefulness must be taken"

---

## Round 0.2 · Minor Revisions

Please address the comments of Reviewer 1

Reviewer 1 ·

Basic reporting

The same as my previous report, as the manuscript was not significantly changed.

Experimental design

The same as my previous report, as the manuscript was not significantly changed.

Validity of the findings

The same as my previous report, as the manuscript was not significantly changed.

Additional comments

The authors unfortunately did not improve the quality of the article regarding any of my three main concerns:
1 - There are known issues regarding the quality of the dataset, that have been noticed by other groups that have given up on using this dataset. For example, the positive class for covid-19 is not expected to be 40 times more prevalent than the negative (as in the case of HSL). The authors need to need to make this clearer throughout the manuscript, especially in the conclusion.
2- The authors should perform statistical analysis to compare whether the variables have similar distributions for the three centers, regardless of the outcome. All that the authors have provided in the manuscript are visual comparisons according to the classes.
3- The authors need to use the area under the ROC curve (AUROC) the main metric, as it is the standard in machine learning studies. It is fine to add the other ones for specific analyzes, but the AUROC should be the main metric to allow for comparisons with other studies.

·

Basic reporting

no comment

Experimental design

no comment

Validity of the findings

no comment

Reviewer 3 ·

Basic reporting

The authors have addressed all points raised by this reviewer.

Experimental design

No comment.

Validity of the findings

No comment.

Additional comments

The authors have addressed all points raised by this reviewer (as well as those raised by two other reviewers). I consider the revised manuscript adequate for publication.

---

## Round 0.3 · accepted · Accept

Authors addressed the reviewer 1's concern satisfactorily. I thus recommend the acceptance of the manuscript.